# Structure-guided inhibition of the cancer DNA-mutating enzyme APOBEC3A

Stefan Harjes[1], Harikrishnan M. Kurup [1], Amanda E. Rieffer[2], Maitsetseg Bayarjargal [1,6], Jana Filitcheva[1], Yongdong Su [1,7], Tracy K. Hale [1], Vyacheslav V. Filichev [1,3] ✉, Elena Harjes [1,3] ✉, Reuben S. Harris [4,5] ✉ & Geoffrey B. Jameson [1,3] ✉

The normally antiviral enzyme APOBEC3A is an endogenous mutagen in human cancer. Its single-stranded DNA C-to-U editing activity results in multiple mutagenic outcomes including signature single-base substitution mutations (isolated and clustered), DNA breakage, and larger-scale chromosomal aberrations. APOBEC3A inhibitors may therefore comprise a unique class of anti-cancer agents that work by blocking mutagenesis, slowing tumor evolvability, and preventing detrimental outcomes such as drug resistance and metastasis. Here we reveal the structural basis of competitive inhibition of wildtype APOBEC3A by hairpin DNA bearing 2′-deoxy-5-fluorozebularine in place of the cytidine in the TC substrate motif that is part of a 3-nucleotide loop. In addition, the structural basis of APOBEC3A's preference for YTCD motifs (Y = T, C; D = A, G, T) is explained. The nuclease-resistant phosphorothioated derivatives of these inhibitors have nanomolar potency in vitro and block APOBEC3A activity in human cells. These inhibitors may be useful probes for studying APOBEC3A activity in cellular systems and leading toward, potentially as conjuvants, next-generation, combinatorial anti-mutator and anti-cancer therapies.

The APOBEC3 family of enzymes (A3A-A3H) forms part of the innate immune response against viruses and transposons[1–3]. These enzymes deaminate cytosine to uracil in single-stranded (ss)DNA[4–7] and, a subset, also in RNA[8–10]. Two family members, A3A and A3B, have also been implicated in deaminating genomic DNA cytosines, which can result in mutations that fuel tumor development and contribute to poor disease outcomes including drug resistance and metastases[11–14]. Recent studies have also shown that A3A contributes to the overall constellation of APOBEC signature mutations in human cancer cell lines[15,16] and, importantly, is capable of causing carcinogenesis in mice[17,18]. A3

inhibitors have been proposed as a therapeutic strategy to prevent A3-mediated evolution of primary tumors into lethal metastatic and drug-resisting secondary growths[19,20].

A3A exhibits an intrinsic preference for deamination of cytosine bases within 5′-YTCD motifs, where Y denotes C or T and D is A, G, or T[15–17,21–26]. The strong preference for TC dinucleotide motifs is explained by crystal and NMR structures of linear ssDNA bound to an inactive mutant of A3A, which revealed the atomic contacts (3 hydrogen bonds) with thymine (T⁻¹) as well as the binding pocket for the target cytosine[27–33]. However, preferences of A3A for nucleotides

[1]School of Natural Sciences, Massey University, Palmerston North, New Zealand. [2]Department of Biochemistry, Molecular Biology, and Biophysics, University of Minnesota–Twin Cities, Minneapolis, MN, USA. [3]Maurice Wilkins Centre for Molecular Biodiscovery, University of Auckland, Auckland, New Zealand. [4]Department of Biochemistry and Structural Biology, University of Texas Health San Antonio, San Antonio, TX, USA. [5]Howard Hughes Medical Institute, University of Texas Health San Antonio, San Antonio, TX, USA. [6]Present address: Department of Biochemistry, University of Washington, Seattle, WA, USA. [7]Present address: Department of Pediatrics, Emory University School of Medicine, and the Aflac Cancer and Blood Disorders Center, Children's Healthcare of Atlanta, Atlanta, GA, USA. ✉e-mail: v.filichev@massey.ac.nz; e.harjes@massey.ac.nz; rsh@uthscsa.edu; g.b.jameson@massey.ac.nz

flanking the T<u>C</u>-sequence are as-yet-unexplained. Crystallographic studies also revealed that, upon binding to A3A, linear ssDNA substrates adopt a distinctive *U*-shape that projects cytosine into the A3A active site[27,28]. Consistent with this observation, DNA hairpins with loops of 3- and 4-nucleotides (nt) have been shown to be more potent substrates of A3A in comparison to linear analogs[16,34–38]. Importantly, DNA hairpins are both physiologically and pathologically relevant. Such structures are ubiquitous in nature, especially at inverted repeats, which can cause stalling of DNA replication and transcription complexes, genomic instability, and a general predisposition to mutagenesis[38,39].

Prior studies have shown that linear ssDNA substrates with 2′-deoxyzebularine (**dZ**) and 5-fluoro-dZ (**FdZ**) in place of the target cytidine (C) are weak inhibitors of A3A[40–42], and that these cytidine analogs as free nucleosides are non-inhibitory[42]. Additional work by our group and others has shown greater inhibition of A3A in vitro with *U*-shaped oligonucleotides containing **dZ, FdZ**, or 5-methyl-2′-deoxyzebularine transition-state trapping molecules[43–45], which agrees well with biochemical studies comparing linear and hairpin substrates and systematically varying hairpin stems and loops[16,26,34,35]. Here, we use

X-ray crystallography to determine high-resolution structures of wildtype A3A/hairpin-inhibited complexes and demonstrate the underlying mechanism of inhibition. Importantly, these structures explain why 3-nt hairpin loops with TTC (or TTFdZ) are preferred A3A substrates (and inhibitors). Larger 4-nt loops extrude one nucleotide to optimally fit around the A3A active site. Moreover, here we also demonstrate that an FdZ-hairpin, TTFdZ-hairpin, is not only a potent nanomolar inhibitor in vitro in biochemical assays but is also capable of inhibiting wildtype A3-catalyzed chromosomal DNA editing in living cells.

## Results

### Deamination and inhibition mechanisms are conserved

The mechanism for cytosine deamination by cytosine deaminases (CDA) was proposed based on crystal structures in which zebularine and 5-fluorozebularine accept a $Zn^{2+}$-bound water molecule across the N3-C4 double bond forming a tetrahedral intermediate in complex with the enzyme[46–48] (Fig. 1a). Accordingly, our prior work with A3A has indicated that, of these two cytosine analogs, the latter fluoro-containing nucleobase is a more potent inhibitor in the context of

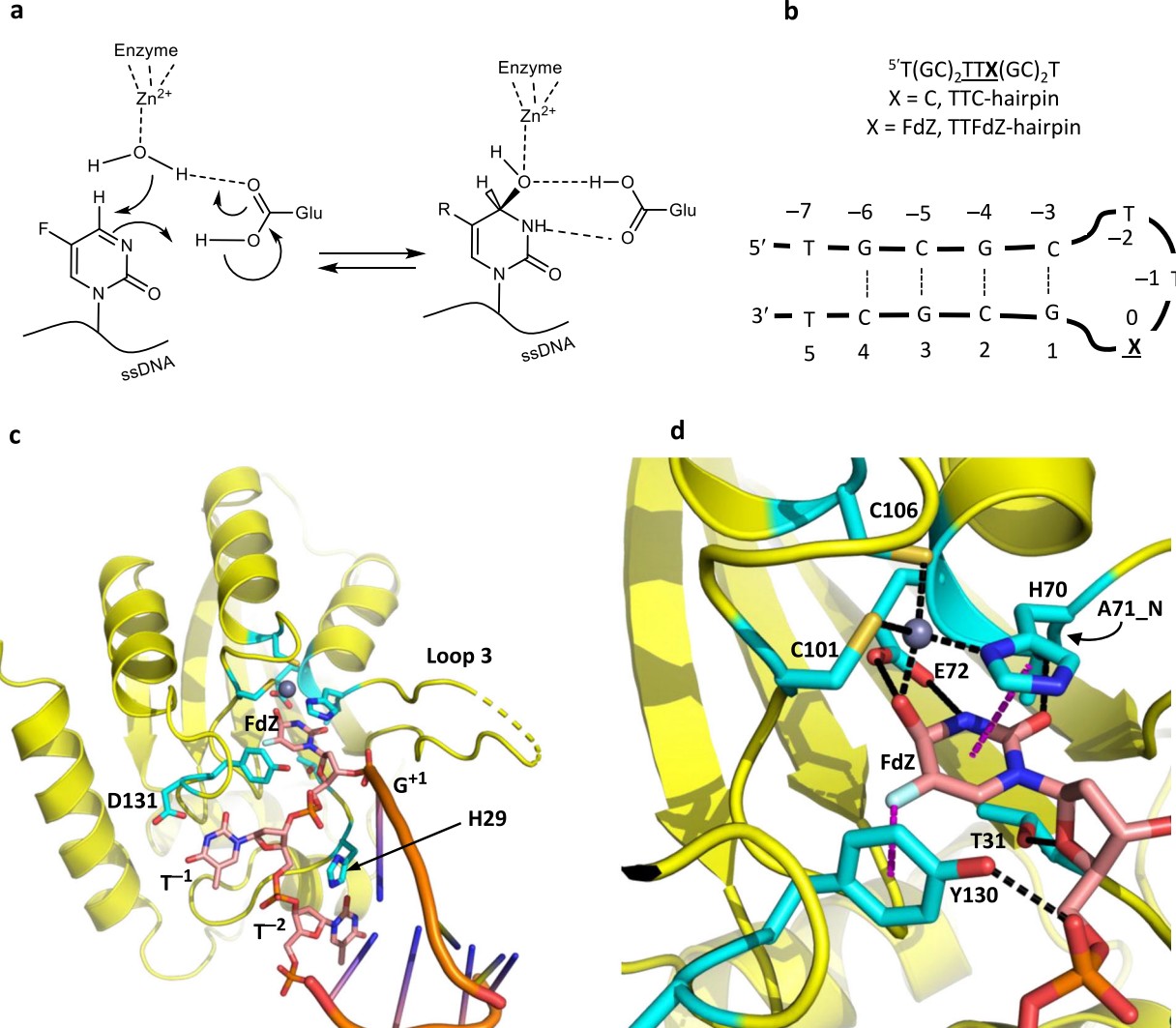

**Fig. 1 | Inhibitor 5-fluoro-2′-deoxyzebularine (FdZ) embedded in hairpin DNA is hydrolyzed to form a tetrahedral species coordinated to $Zn^{2+}$ of wildtype A3A.** **a** Schematic of reaction of FdZ with water activated by $Zn^{2+}$, analogous to that of cytidine/cytosine deaminases, showing critical role of general acid-base Glu72. **b** Schematic of TTFdZ- and TTC-hairpins. **c** Structure of wildtype A3A (yellow) in complex with TTFdZ-hairpin inhibitor (orange). Carbon atoms of nucleotides of the TTFdZ loop are shown in salmon-pink, ligands to the $Zn^{2+}$ center and key protein residues interacting with TTFdZ-hairpin in cyan. **d** Zoomed-in structure of wildtype A3A with TTFdZ-hairpin inhibitor highlighting the hydrolyzed $FdZ^0$ coordinated to the active-site $Zn^{2+}$ and hydrogen bonding to Glu72 (cyan).

ssDNA[41]. Therefore, to address whether A3A utilizes the same deamination mechanism, we crystallized wildtype A3A complexed with an inhibitory hairpin (TTFdZ-hairpin: 5′-T(GC)$_2$TTFdZ(GC)$_2$T; Fig. 1b, c). Two different crystal forms were obtained and resolved to 2.80 and 2.94 Å, with slightly different packing in the unit cell of the crystallographically independent pairs of molecules, but close superposition of each molecule of the pair (superpositions are provided in Supplementary Fig. S1a, b). A3 enzymes in general prefer ssDNA over RNA as substrates and, accordingly, all deoxyribose moieties bound by A3A adopt the standard DNA C2-*endo* conformation in this crystal structure and in other structures described below, which helps to determine the positioning of T$^{-1}$ and C$^0$ (or FdZ$^0$) into the −1 and target nucleobase binding pockets, respectively. The loop region, anchored by the target cytosine C$^0$ and thymine T$^{-1}$, is bound to wildtype A3A in a *U*-shaped conformation as reported for linear ssDNA bound to a Glu72-to-Ala catalytic mutant[27,28], but with several important differences discussed below in subsequent sections.

Mass-spectrometric characterization of DNA hairpins and X-ray structural information of this and other hairpin structures in complex with A3A are provided in Supplementary Table S1 and S2. Supplementary Fig. S1c highlights similarities and differences of the binding of C$^0$ and FdZ$^0$ into the active site. Evidence for hairpin structure in solution is shown with representative CD and NMR spectra in Supplementary Fig. S1d.

Zooming into the target nucleotide, FdZ$^0$, in both structures, the N3-C4 double bond of FdZ is hydrolyzed so that the hydroxy group at C4 represents the tetrahedral intermediate in the deamination reaction (Fig. 1a, c, d; additional view in Supplementary Fig. S1c). The FdZ$^0$ exhibits *R*-stereochemistry, 4-(*R*)-hydroxy-3,4-dihydro-2′-deoxy-5-fluorozebularine. The hydroxyl group at C4 is coordinated to the Zn$^{2+}$ center and is derived from the water/hydroxide ion bound to the Zn$^{2+}$ in the substrate-free state. Tetrahedral coordination at the Zn$^{2+}$ is completed by His70, Cys101, and Cys106. The catalytic glutamic acid residue, Glu72, which functions as a general acid/base, hydrogen bonds to C4-OH and N3-H of hydrated FdZ. We expect for such interactions that Glu72 is present in the carboxylate form in the crystal structure. The 5-fluoro group of FdZ is accommodated comfortably, abutting Tyr130 (illustrated in Fig. 1d). These results with wildtype A3A, together with prior work indicating hydration of dZ and its derivatives in structures with CDA and A3G, combine to demonstrate a universal mechanism of target nucleobase engagement, deamination, and inhibition[27,28,48–51].

## A3A binds similarly to 3- and 4-nt DNA hairpin loops

To ascertain generality of the interactions observed between hairpin substrates and A3A, we determined crystal structures of A3A as its inactive E72A mutant in complex with three distinct DNA hairpins with loop regions of either 3- or 4-nt (depictions in Fig. 1b and Fig. 2a). All structures exhibit a similar crystal packing (Supplementary Fig. S2) and a common tertiary structure for the protein (with minor changes on absence of Zn$^{2+}$; Supplementary Fig. S3a, b). Surprisingly, they also share very similar conformation of the hairpin loop region's TT(C/FdZ)G moiety (Fig. 2b, c). The superpositions of A3A-E72A in complex with the 3-nt loop, TTC-hairpin (at 2.22 Å resolution), and the 4-nucleotide loops, ATTC-hairpin (at 1.91 Å resolution) and CTTC-hairpin (at 3.15 Å resolution), are mostly similar with an average RMSD of 0.43 Å (ribbon schematics in Fig. 2b and Supplementary Fig. S3c). However, an unexpected difference emerged between 3- and 4-nt hairpin structures with the clear flipping-out of the nucleotide at position −3 of the 4-nt loop, with the next nt, C$^{-4}$, maintaining hydrogen bonding with G$^{+1}$, as in structures with a three-nt loop (Fig. 2c).

Apart from extrusion of the 5′ nucleotide of 4-nt loops, the structures of wildtype A3A (two near-isoforms) and A3A-E72A, along with their respective hairpin DNA substrates, all superimpose very closely (Fig. 2d, e; Supplementary Fig. S1c). This close superposition

indicates that the catalytic Glu72-to-Ala substitution has negligible effect on tertiary structure. This conclusion is supported by wildtype A3A and its catalytic mutant derivative E72A exhibiting similar thermostabilities[52]. In all instances, the C$^0$ and T$^{-1}$ of TTC-hairpin bind to inactive A3A-E72A identically to the binding of FdZ$^0$ and T$^{-1}$ in TTFdZ-hairpin to wildtype active A3A (Fig. 2d, e; Supplementary Fig. S1c). For both FdZ$^0$ and dC$^0$ the carbonyl oxygen at C2 hydrogen bonds to the NH of Ala71 and the nucleobase base stacks with Zn$^{2+}$ ligand His70 and makes an edge-to-face π interaction with Tyr130. The NH$_2$ substituent on C$^0$ hydrogen bonds to the peptide carbonyl of Ser99 and to a water molecule. This water is located similarly to the Glu72 carboxylate oxygen of wildtype A3A that hydrogen bonds to the OH at C4 of hydrolysed FdZ$^0$ (Fig. 2e; Supplementary Fig. S1c).

## Structural explanation for hairpin loop preference of A3A

Thymine T$^{-1}$ and cytosine C$^0$ of the TTC-hairpin substrate bind identically to that observed crystallographically for binding of linear ssDNA to A3A-E72A[27–30] (Fig. 3a, b). As reported for other structures of A3A-E72A in complex with ssDNA[27,28], specificity for thymine at position −1 is defined by hydrogen bonding to the peptide NH and carboxylate group of Asp131 (Fig. 3c). However, the binding of T$^{-1}$ and C$^0$ does not explain the fact that hairpin loop cytosines are preferred substrates for A3A compared to linear substrates. In contrast to existing structures of linear ssDNA with mutant A3A constructs where nucleotides other than T$^{-1}$, C$^0$ and W$^{+1}$ (W = A, T) are absent or poorly defined in electron-density maps[27,28,32,33], our structures reveal this information and provide a molecular basis for higher reactivity of hairpin DNA substrates and the corresponding enhanced potency of hairpin-based inhibitors (Fig. 3).

First, His29 base-stacks with G$^{+1}$, and T$^{-2}$ base-stacks with a pyrimidine in the stem (C$^{-3}$ for the 3-residue loop or C$^{-4}$ for the 4-residue loop) (Fig. 2c, e; Fig. 3b, d, e). Second, the positively charged guanidinium moiety of Arg28 forms a cation-π interaction with the nucleobase T$^{-2}$, thereby further stabilizing T$^{-2}$ placement for interaction with His29 (Fig. 3d). Arg28 also forms a hydrogen bond to an oxygen atom of the phosphate linking A$^{-3}$/C$^{-3}$ to T$^{-2}$, and the cytosine at position −4 is in register to hydrogen bond with guanine at position +1 as a part of hairpin's stem (Fig. 3d). Third, the tight turn between the nucleotide at position +1 and the target cytosine at position 0 is stabilized by a bifurcated hydrogen bond between Nδ$_1$ (amine tautomer) of His29 and O4′ of 2-deoxyribose at +1 and an oxygen atom of the phosphate group that links nucleotides C$^0$ and T$^{-1}$ (Fig. 3e). This tight turn to project C$^0$ into the active site is accomplished with non-standard torsional angles for the phosphate groups, relative to expected values for A- or B-form DNA, as detailed in Supplementary analysis and discussion. Last, the peptide NH group of Lys60 hydrogen bonds to the phosphate linking nucleotides at positions 0 and +1, and the -NH$_3^+$ moiety of its side chain forms a salt bridge to the phosphate group linking nts +1 and +2 (Fig. 3e). Taken together, the hairpin stem bestows restricted conformational flexibility for the A3A-binding loop region of hairpin DNA compared to linear ssDNA, thereby enabling enhanced interactions with A3A loop 1 residues Arg28 and His29. Supplementary Fig. S4 shows a space-filled representation that highlights the tight packing of the TTC-hairpin into the A3A active-site cavity and against loop 3, along with a complete depiction of the base-pairing of the stem of TTC-hairpin and the atoms in hydrogen bonding and van der Waals contact.

The close superposition of the stems, which include in several structures two AT pairs, along with GC pairs, and the lack of specific interaction of most of the DNA stem with the protein (Fig. 2b, c; Supplementary Fig. S4a), suggests that reactivity of dC-containing hairpins and inhibition by **FdZ-** (or **dZ-** or 5-methyl-dZ-) hairpins is unrelated to stem composition of DNA hairpins, where substrate or inhibitor moiety is located at the 3′ end of the loop. This is consistent with the recent observation that variation of stem composition had negligible effect on inhibition[44].

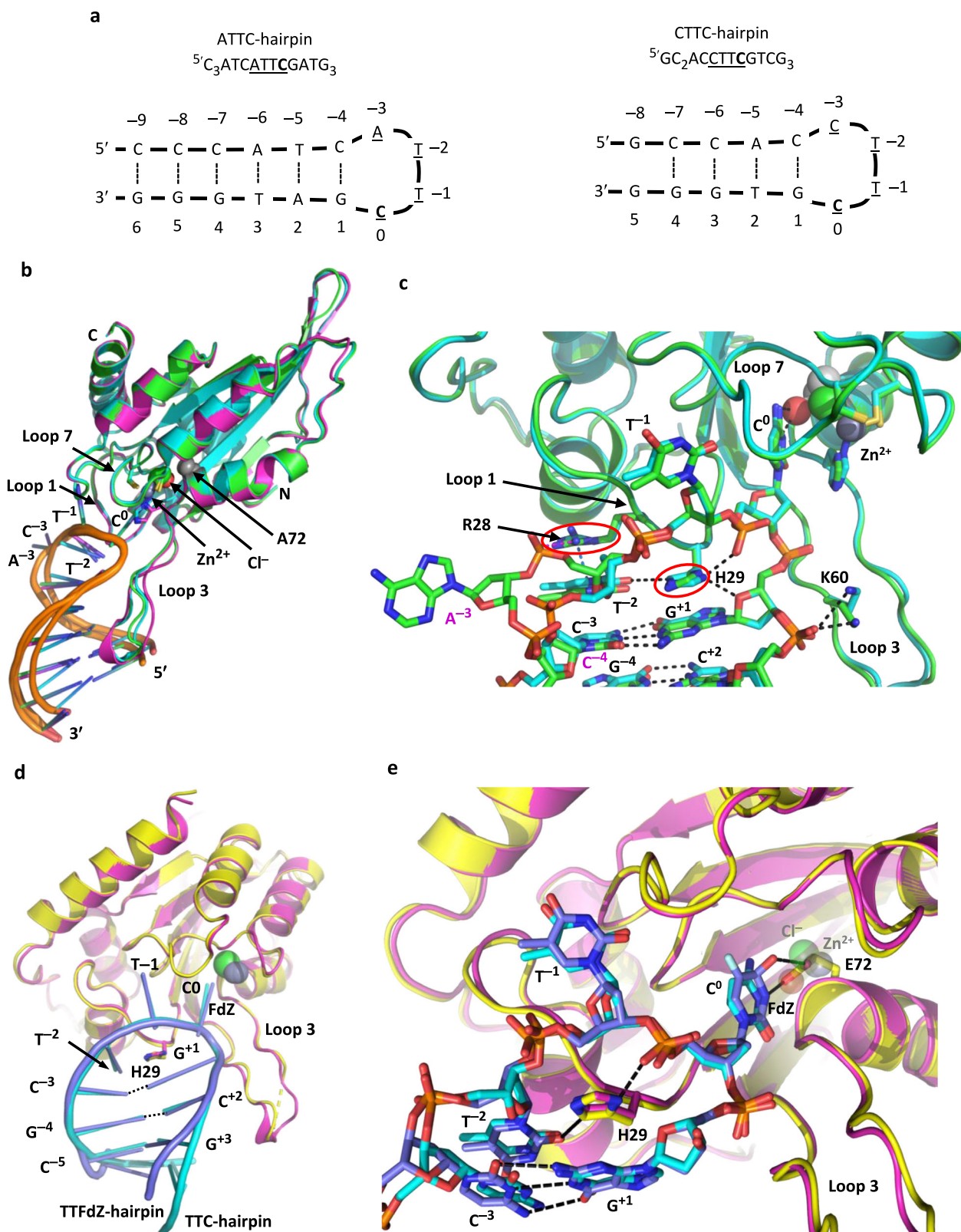

## Structural explanation for A3A's −2 pyrimidine preference

The composite mutation spectra of human A3A in multiple model systems have revealed a marked bias for a pyrimidine (C or T) at nucleotide position −2[16,17,24,25]. In addition to roles of His29 and Asp131 in positioning cytosine in the substrate-binding pocket and dictating the strong preference for thymine at position −1, as detailed in the previous section, His29 also helps to determine the preference of A3A for a pyrimidine (T or C) at position −2. By virtue of chemical structures, purines (adenine and guanine) are larger than pyrimidines, and modelling indicates that they cannot be accommodated in the −2 binding site in either a *syn* conformation of the glycosidic bond (where there is repulsive interaction of imine N3 with the phosphate group linking nucleotides −2 and −3) or an *anti* conformation of the glycosidic bond (Supplementary Fig. S5). In the *anti* conformation, purines

**Fig. 2 | Stereochemistry of interactions of wildtype A3A and A3A-E72A with hairpin DNA featuring 3- and 4-nucleotide loops is conserved. a** Sequences of hairpins with 4-nt loops: ATTC- and CTTC-hairpin. **b** Superposition of structures of A3A-E72A in complex with 3-nt loop TTC-hairpin (cyan) and 4-nt loop ATTC-hairpin (green) and CTTC-hairpin (magenta), showing conservation of protein tertiary structure and loop conformation. **c** Zoom into the active-site region of A3A-E72A in complex with TTC-hairpin (cyan) and ATTC-hairpin (green), showing overall binding conservation (structure and hairpin stem orientation) and extrusion of A at position −3 for the 4-nucleotide loop. Red ellipses highlight His29 and Arg28,

which have key roles in determining the conformation of the loop that makes hairpins better substrates for wildtype A3A than corresponding linear ssDNA. **d** Superposition of wildtype A3A (yellow) with TTFdZ-hairpin (grey-blue) and A3A-E72A (magenta) with TTC-hairpin (cyan). **e** Zoom-in (with slight reorientation) of the superposition of wildtype A3A (yellow) with TTFdZ-hairpin (grey-blue) onto A3A-E72A (magenta) with TTC-hairpin (cyan). The water (faded red sphere) and chloride ion (faded green sphere) of A3A-E72A superimpose approximately onto the carboxylate oxygen atoms of Glu72. Key hydrogen bonds are shown with black dashed lines.

lack the C = O moiety at C2 of T or C to form the crucial non-classical hydrogen bond with His29 (Fig. 2c). In addition, the water molecule that bridges the carbonyl O2 of $T^{-2}$ (and potentially also of C at −2) to the peptide backbone NH of His29 cannot be accommodated for purine at −2 (Fig. 3d, Supplementary Fig. S5b, d). Therefore, this quasi-base pair between pyrimidine at position −2 and His29 is able to stack on top of the first CG base pair (at positions −3 or −4 and +1) at the head of the stem (Fig. 3d).

The importance of Arg28 and His29 to recognition of $T^{-2}$ is shown also by the following results. For A3A on mutation of Arg28 to alanine, diminished activity towards linear ssDNA was reported[27]. Moreover, nutation of His29 to arginine in A3A (H29R) caused a 10-fold diminution of activity against a linear ssDNA compared to wildtype A3A[53,54]. Based on co-crystal structures here, we predict that the longer side chain of Arg in the H29R mutant will place the polar head group in a suboptimal position beyond the reach of O2 of $T^{-2}$ and with poor π-stacking with the nucleobase at +1. Interestingly, the nearly identical (>90%) A3B catalytic domain lacks a corresponding histidine in its loop 1 region (it naturally has an arginine), and A3B shows no preference for pyrimidines in the −2 position of linear ssDNA substrates with target C at position 0[16,55]. As yet, there is no structural information on wildtype A3B$_{CTD}$ with loop 1 in an open conformation with substrate or inhibitor bound to test this supposition.

**Hairpin structures also help explain +1 preference for D (D ≠ C)**
Cytosine at +1 is rarely observed in the A3A-induced mutation spectra in model systems[16,17,24,25] and it is also strongly under-represented in the overall A3 mutation signature in tumors[56,57]. At least for hairpin structures, π-π stacking of His29 with the six-membered ring of $G^{+1}$ (or alternatively $A^{+1}$) and van der Waals interaction of the CH$_2$ group (Cβ) of His29 with the five-membered ring of $G^{+1}$ (or alternatively $A^{+1}$) explains the preference for purines over pyrimidines in the +1 position (Fig. 3d, e). The preference for thymine over cytosine is more subtle. In part, cytosine lacks the electron-donating methyl group of thymine, which leads to less favorable π-π interactions with His29. In addition, modelling T at position +1 and adjusting the Lys30 side chain to a favoured conformation reveals a small hydrophobic pocket that brings the methyl groups of $T^{+1}$ and Ala59 and the methylene groups Cγ and Cε of Lys30 into van der Waals contact. Moreover, the terminal amino group of Lys30 hydrogen-bonds with the carbonyl moiety at C4 of $T^{+1}$ (Fig. 3f); cytosine lacks this carbonyl, instead having an amino group. In this context, we also note that in vitro A3A prefers to deaminate the highlighted C of a suboptimal linear 5′-ATTCC**C**AATT substrate, whereas A3B$_{CTD}$ attacks the 5′-most C[40,54]. Cytosine lacks this methyl group and carbonyl group and thus substrates presenting YTT**C**D (D = A, G, T) to A3A are favoured over those presenting YTT**C**C.

**Hairpin optimization for cellular experiments**
The thermodynamic properties of the TTC hairpin were assessed in solution by CD and NMR and confirmed stable (Supplementary Fig. S1d). Isothermal titration calorimetry (ITC) measurements established that binding is largely enthalpically driven (Supplementary Table S3; Supplementary Fig. S6a, b). Real-time C-to-U deamination

experiments using NMR spectroscopy demonstrated that the TTC-hairpin is a preferred substrate of wildtype A3A in comparison to a linear TTC substrate, which agrees with prior reports using a fluorescence-based assay[16,35,38] (Fig. 4a, b). The kinetic parameters derived from NMR experiments using the integrated Michaelis-Menten equation and Lambert's W function[58] also confirmed that a TTFdZ-hairpin is a 21-fold more potent than a TTFdZ linear ssDNA in blocking wildtype A3A activity (Fig. 4a, b; Supplementary Fig. S6c). A dramatic change in $K_m$ from 3.0 ± 0.9 mM for linear ssDNA to 31 ± 6 μM for TTC-hairpin is the major contributor to a 42-times more efficient deamination of TTC-hairpin if $k_{cat}/K_m$ are compared at pH 7.4 (Supplementary Table S4a). Moreover, the nanomolar potency ($K_i$) of TTFdZ-hairpin inhibitor was not significantly altered by replacing the hairpin phosphate groups with phosphorothioate (PS) linkages (Fig. 4b). Nuclease resistance was confirmed by treating with snake venom phosphodiesterase, which is a strong 3′-exonuclease commonly used to evaluate the stability of oligonucleotides with therapeutic potential[59] (Fig. 4c). Nuclease resistance and inhibitory activity were also shown using A3A-expressing 293 T cell lysates, where the PS-TTFdZ hairpin exhibited stronger inhibitory activity in comparison to a linear PS-TTFdZ ssDNA or a PS-TTT-hairpin negative control (Fig. 4d).

**PS-TTFdZ-hairpin inhibits A3A editing *in cellulo***
To assess stability and localization in living cells, the modified PS-hairpins possessing **dZ** or **FdZ** were fluorescently labelled at the 3′-end (6-FAM). The MCF-7 breast cancer cell line was transfected with these hairpins using the Xtreme GENE™ HP transfection agent. After 18 h these FAM-labelled PS-hairpins were found to localize to the nucleus in a concentration-dependent manner (Fig. 5a; Supplementary Fig. S7). Moreover, the metabolic activity of MCF-7 and another breast cancer cell line MDA-MB-453 was reduced less than 2-fold (Supplementary Fig. S8a).

*In cellulo* inhibition of A3A activity was quantified using a base-editing system reported previously[60,61], except here A3A is detached to the Cas9 nickase complex (Methods). Briefly, A3A-catalyzed editing of a single target cytosine nucleobase within a ssDNA R-loop created by a nuclease-deficient Cas9-guide RNA complex in an *eGFP* reporter construct results in a dose- and time-dependent restoration of eGFP fluorescence (Methods; Supplementary Fig. S8b). This reporter was stably integrated into the chromosomal DNA of 293 T cells to mimic an R-loop environment that A3A might encounter in cancer cells. Upstream of the mutated *eGFP* codon lies a linked wildtype *mCherry* gene for calculating the efficiency of base editing (eGFP + /mCherry + ). With PS-TTT-hairpin as a A3A-non-binding control, little change in generation of eGFP fluorescence over time was observed as a function of concentration (Supplementary Fig. S9). However, in the presence of increasing concentrations of PS-TTFdZ-hairpin inhibitor, there was marked suppression of the generation of fluorescence (Fig. 5b). The maximum rate of editing by A3A was 2.4-fold lower in the presence of 7.5 μM inhibitor than in the absence of inhibitor, suggesting an upper limit on IC$_{50}$ of ~5 μM (Fig. 5c). The difference in cell viability between PS-TTT and PS-TTFdZ hairpins was minimal in comparison with A3A inhibition data for these oligos at the same concentrations in 293 T cells (Supplementary Fig. S8a).

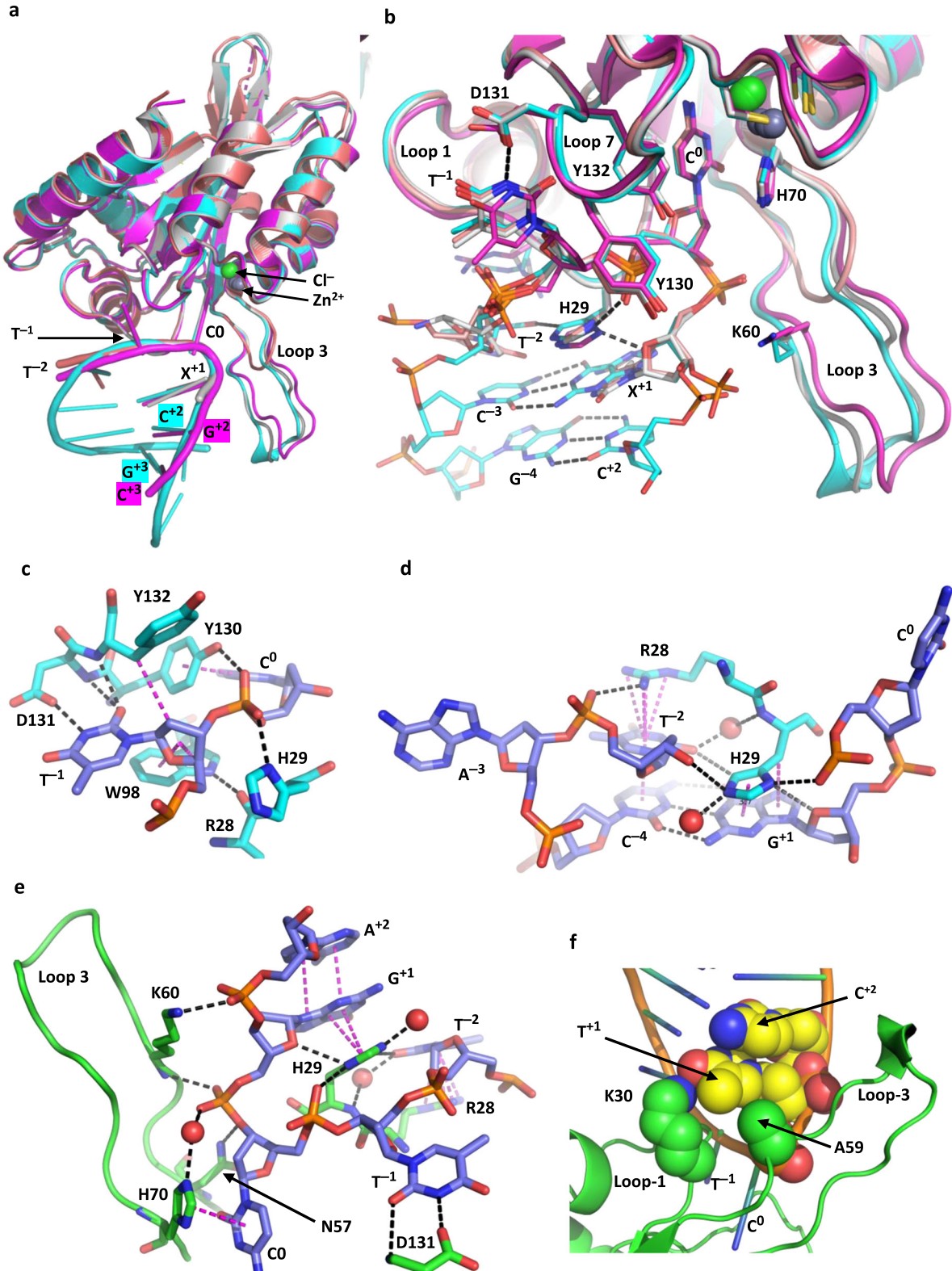

## Discussion

Our structural studies establish that the stem-loop preconfigures the primary TC recognition motif in optimal position for binding to A3A, such that hairpin DNAs are more reactive substrates and as the 2′-deoxy-5-fluorozebularine derivative is a more potent inhibitor of A3A than linear ssDNA. Both 3- and 4-nt loops present the TC or TTFdZ motif in an identical configuration to A3A. Although hinted at in earlier structures with ssDNA[24], a crucial role for A3A His29 (and to a lesser

extent Arg28) in substrate binding is demonstrated here and, importantly, also shown to help explain its preference for deamination of YTCD motifs (Y = C, T; D = A, G, T).

Although we have not specifically tested the hairpin inhibitors described here against related human enzymes, the most similar TC-preferring A3 family member (A3B) is not known to prefer hairpin substrates[16,55,62] and other TC-preferring A3s have yet to be tested systematically with hairpins. A3G and AID, with non-TC preferences,

**Fig. 3 | Key interactions of hairpin and linear DNA with A3A-E72A that define recognition of the TC motif and *U*-shaped conformation and preferences at −2 and +1 positions. a** Superposition of a representative A3A-hairpin complex (E72A-TTC hairpin; cyan) and A3A-E72A-Cys171A with 15-nt linear ssDNA (5keg; grey), A3A-E72A with 15-nt linear ssDNA (5sww; magenta)[27], and A3Bctd-E255A-AL1swap-QM-ΔL3 with 7-nt linear ssDNA (5td5; salmon-pink). Only underlined nucleobases are visible in these ssDNA complexes: 5′-T$_7$TCTT$_5$ (5keg), 5′-A$_6$ATCGGGA$_3$ (5sww), and 5′-T$_2$TTCAT (5td5). Zn$^{2+}$ is shown as grey spheres and Cl$^-$ as green spheres. **b** Zoom-in showing details of shared interactions of TCX (X = G, A, T) of hairpin and linear DNA. DNA passes over loop 1 (bearing Arg28 and His29) and under loop 7 (bearing Tyr130, Asp131, and Tyr 132). Asp131 largely specifies binding of T at position −1; Tyr130 makes an n-π* interaction with the phosphate linking nucleotides −1 and 0. Loop 3 contacts DNA at metal-ligand His70 and at Lys60. G for TTC-hairpin and A for 5′-T$_2$ TTCA$T$ (at position +1) overlap poorly, as do T at −2 (observed only for TTC-hairpin and 5′-$T_2$ TTCA$T$ (5td5)). Zn$^{2+}$ is shown as grey spheres and Cl$^-$ as green spheres. **c** Details of interactions of T$^{-1}$ with Asp131 and the cluster of aromatic residues. **d** Details of interactions of Arg28 and His29 of loop 1 of A3A-E72A with the ATTC-hairpin. Key hydrogen-bonding interactions are shown by black dashes. Cation-π interactions between Arg28 and T$^{-2}$, π-π interactions between T$^{-2}$ and C$^{-4}$ and between His29 and G$^{+1}$, and dispersion interactions are shown as magenta dashes. Water molecules are shown as red spheres. **e** Details of interactions that mediate the tight turn between C$^0$ and G$^{+1}$. Peptide NH and amino side chain of Lys60 make key contacts with ATTC-hairpin. **f** Thymine modelled at position +1 (in place of G) for TTC-hairpin bound to A3A-E72A. In addition to the hydrogen bond of the terminal amino group of Lys30 to the carbonyl at C4 (overlapping blue and red spheres), there are van der Waals contacts made by the methyl group of T$^{+1}$ with the methyl group of Ala59 and the methylene groups Cγ and Cε of Lys30 modelled in a highly preferred rotamer.

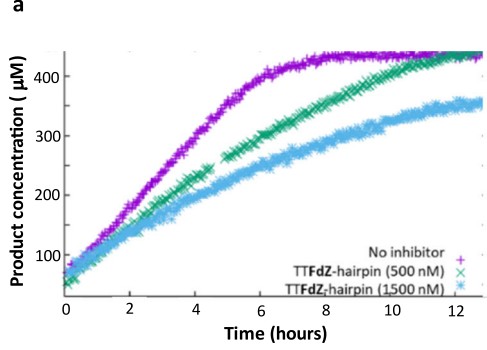

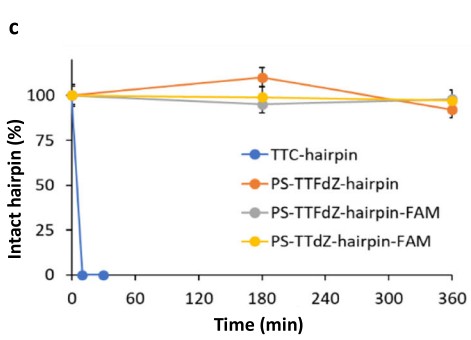

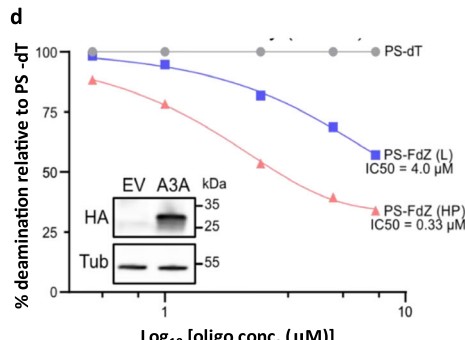

| Protein | Oligo | $K_i$ (/nM) |
|---|---|---|
| Wt A3A | FdZ-linear | 2400 ± 900 |
| Wt A3A | TTFdZ-hairpin | 117 ± 15 |
| Wt A3A | PS-TTFdZ-hairpin | 160 ± 70 |

**Fig. 4 | TTFdZ-hairpin and its nuclease-resistant phosphorothioated derivative are potent inhibitors of A3A compared to linear ssDNA with FdZ. a** Plot of product concentration *versus* time in the absence and presence of TTFdZ-hairpin at 500 and 1500 nM; concentration of A3A 140 nM; concentration of TTC-hairpin substrate 500 μM. **b** Table shows derived inhibition constants, $K_i$ for linear FdZ ssDNA (A$_2$T$_2$FdZA$_4$), TTFdZ-hairpin and its phosphorothioated analog PS-TTFdZ-hairpin. The kinetic parameters were derived from the integrated Michaelis-Menten equation by means of Lambert's W function[58], which provides more robust estimates for kinetic parameters $K_m$ and $V_{max}$ than analysis of initial rates only. **c** Percentage of intact hairpins (15 μM) over time upon treatment with snake venom phosphodiesterase (phosphodiesterase I, Sigma, 32 mU/mL) in 50 mM Tris-HCl buffer, 10 mM MgCl$_2$, pH 8.0 for the indicated times at 37 °C. Data are presented as mean values of two independent experiments on each sample; errors bars are estimated instrumental error of ± 5%. Full experimental details are available in Supplementary Information. **d** Concentration-dependent inhibition of A3A in cell lysates by phosphorothioated TTFdZ-hairpin. Bands from SDS-PAGE (Supplementary Fig. S10b) were quantified, normalized to the PS-TTT-hairpin (denoted PS-dT oligo) and used to determine the IC$_{50}$ for linear PS-TTFdZ (denoted PS-FdZ (L)) and PS-TTFdZ-hairpin (denoted PS-FdZ (HP)). This experiment was repeated once more (*n* = 2 biologically independent replicates), and data are from one representative gel. The inset panel is an immunoblot showing expression of A3A-HA from cell lysate. A3A-HA was detected by an anti-HA antibody while anti-tubulin was used as a loading control. EV denotes the empty vector control. Raw immunoblot images are located in Supplementary Fig. 10a.

are unlikely to be inhibited by the hairpins described here. However, when hairpin or other A3A inhibitors get closer to clinical development, these and other off-target possibilities should be examined in dedicated biochemical and cellular experiments.

A structural understanding of the hairpin preference of A3A helped inform the design of substrate-mimicking FdZ inhibitors. Importantly, phosphorothioated derivatives are resistant to nuclease degradation and can be directed to the nucleus with the aid of commonly used transfection reagents. Moreover, we have obtained an important proof of concept here through the inhibition of the mutagenic activity of A3A in living cells with a PS-FdZ hairpin.

Further optimization of such inhibitors may lead to small molecules that can be used in a therapeutic setting to slow rates of tumor evolution and improve clinical outcomes for patients with A3A-driven tumors.

## Methods

### Oligodeoxynucleotide synthesis and purification

The general strategy for the synthesis of **dZ** and **FdZ** and their incorporation into DNA oligomers has been described by us elsewhere[43]. 3-[(Dimethylaminomethylidene)amino]-3*H*-1,2,4-dithiazole-3-thione (DDTT, Sulfurizing Reagent II from GlenResearch, USA) was used for

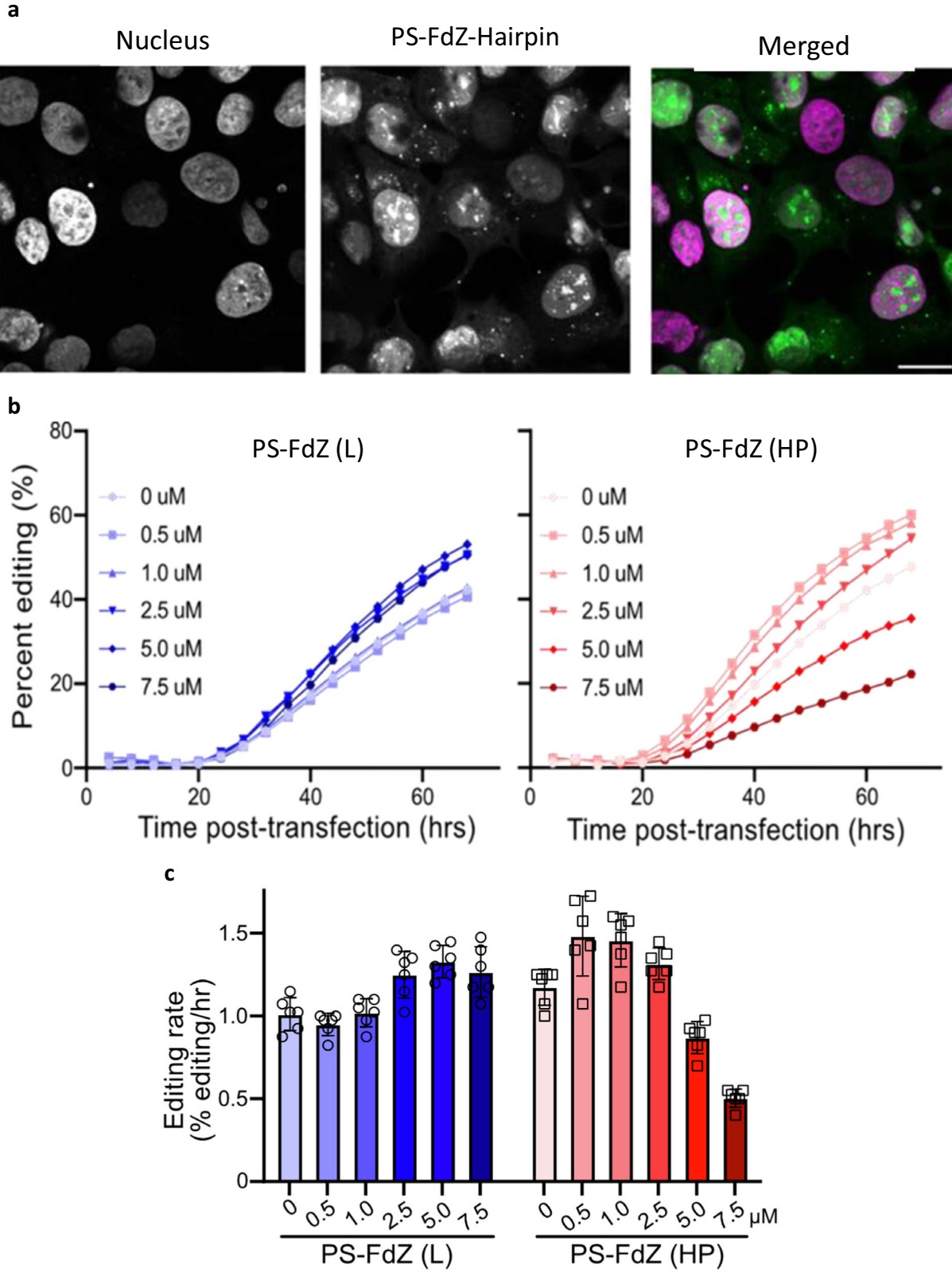

sulfurization of oligos. The sulfurization step (2–4 min) was conducted before capping as a replacement of the standard oxidation step. FAM is located at 3'-end of the oligo and was obtained by synthesizing an oligo on a controlled pore glass loaded with 4-[6-[(2 S, 4 R)-4-hydroxy-2-(DMT-O-methyl)pyrrolidin-1-yl]-6-oxohexyl]carbamoylfluorescein purchased from PrimeTech (Cat. number: 008a-500, Minsk,

Belorussia). **dZ** and **FdZ** phosphoramidites were synthesized as previously described in refs. 41–43.

The final detritylated **dZ**-containing oligos were cleaved from the solid support and deprotected at room temperature using conc. $NH_4OH$ overnight. **FdZ**-containing oligos were deprotected on the solid support by a two-step procedure with 10% $Et_2NH$ in $CH_3CN$ for

**Fig. 5 | Fluorescently tagged PS-TT(F)dZ-hairpin-FAM localizes to the cell nucleus where *in cellulo* A3A-editing activity is inhibited by phosphorothioated TTFdZ-hairpin. a** Representative images of asynchronously-grown MCF-7 cells transfected using Xtreme GENE™ HP with either no hairpin (top panel) or 1.25 μM of fluorescently-tagged (6-FAM) PS-TTFdZ-hairpin. MCF-7 cells were incubated for 16 h with hairpin DNA and Xtreme GENE ™ HP. Images have the pseudo-coloured panels overlaid: nucleus (magenta) and PS-TTFdZ-hairpin-FAM (green). Variability in hairpin uptake is attributed to cells being at different stages of their cycle. Scale bars, 20 μm. Additional images may be found as Supplementary Fig. S7. **b** PS-TTFdZ-hairpin (denoted PS-FdZ (HP)) shows concentration-dependent inhibition of A3A-editing activity in comparison with PS-FdZ (L), a linear, fully phosphorothioated oligonucleotide AT₃**FdZ**AT₃. Biological replicates, establishing reproducibility, are shown in Supplementary Fig. S9a, b. Variability in plasmid transfection prevents averaging of results. The transfection reagent (TransIT-LT1), administered at constant concentrations across all *in cellulo* experiments in presence or absence of DNA species, appears to have a slight activating effect on A3A editing activity at low concentrations of both hairpin and linear DNA. **c** Maximum A3A-catalyzed editing rate at various concentrations of PS-FdZ (L) and PS-FdZ (HP). Data points are the rolling slope for each concentration as described in Methods, and representative of three biologically independent experiments ($n = 3$). Replicates are shown in Supplementary Fig. S9a, b. Error bars represent standard deviation from the mean (mean +/− SD).

5 min, followed by incubation of the support in ethylenediamine/toluene mixture (1/1, v/v) for 2 hrs at room temperature[63]. The support was washed with toluene (3 × 1 mL), dried *in vacuo* and the deprotected **FdZ**-containing oligo was released in H₂O (1 mL).

The deprotected oligos in solution were freeze-dried and dry pellets were dissolved in milli-Q water (1 mL) and purified and isolated by i) reverse-phase HPLC on 250/4.6 mm, 5 μm, 300 Å C18 column (Thermo Fisher Scientific) in a gradient of CH₃CN (0 → 20% for 20 min, 1.3 mL/min) in 0.1 M TEAA buffer (pH 7.0) with a detection at 260 nm or ii) ion-exchange (IE) HPLC using TSKgel Super Q-5PW column from TSK in buffer A [25 mM Tris·HCl, 20% CH₃CN, 10 mM NaClO₄, pH 7.4] and buffer B [25 mM Tris·HCl, 20% CH₃CN, 600 mM NaClO₄, pH 7.4]. Gradients: 3.7 min 100% buffer A, convex curve gradient to 30% B in 11.1 min, linear gradient to 50% B in 18.5 min, concave gradient to 100% B in 7.4 min, keep 100% B for 7.4 min and then 100% A in 7.3 min. Flow rate: 0.8 mL/min with a detection at 260 nm.

Oligonucleotides were freeze-dried, pellets were dissolved in milli-Q water (1.5 mL) and desalted by reverse-phase HPLC on a 100/10 mm, 5 μm, 300 Å C18 column (Phenomenex) in a gradient of CH₃CN (0 → 80% for 15 min, 5 mL/min) in milli-Q water with detection at 260 nm. Pure products were quantified by measuring absorbance at 260 nm, analyzed by ESI-MS and concentrated by freeze-drying (Supplementary Table S1).

### Expression and purification of A3A constructs
A3A-E72A was expressed and purified as described in ref. 32. A3A-E72A was used for structural studies and ITC experiments with the substrate. Wildtype A3A, which was recombinantly expressed with the His₆ tag at the C-terminal end in *E. coli* and purified as described in ref. 43, was used for kinetic and structural studies with the inhibitor. The yield of wildtype A3A from 1 to 10 L expression was usually not enough to justify size-exclusion chromatography purification. The protein, both A3A-E72A and wildtype A3A, was transferred from high-salt buffer used for purification (50 mM phosphate buffer pH 6.5, 300 mM Na acetate, 300 mM choline chloride, 1 mM TCEP) into low-salt buffer (1 mM phytic acid pH 7.0, 1 mM NaF, 1 mM NaCl, 1 mM TCEP) for biophysical characterization and crystallization by "washing" 3 times using centrifugal filtration with 10 kDa cut-off.

### Co-crystallization of A3A constructs with hairpin substrates
A3A-E72A (1–4 mM in low salt buffer) was mixed with oligonucleotides (10 mM in TE buffer: 10 mM Tris/HCl pH 7.9, 1 mM EDTA) in a 1:2 molar ratio (protein:ligand) and diluted to 0.75 mM using above low salt buffer. Dilution was done with protein buffer. The mixture was added to crystallization solution in a 1:1 ratio and the mixture was pipetted on siliconized glass disks and sealed on top of a reservoir of crystallization solution for hanging-drop crystallization at 12 °C. The crystallization solution had the following composition: 100 mM bicine at pH 6.6, 200 mM NaCl, 20 mM putrescine, 1 mM TCEP, 1 mM inositol hexaphosphate (phytic acid) and 45% pentaerythritol propoxylate (5/4 PO/OH). The Zn²⁺-free crystals of A3A-E72A with ssDNA were crystallized using A3A-E72A that had been purified in the presence of 1 mM EDTA.

His₆-tagged wildtype A3A was mixed with inhibitor oligonucleotide in a 1:2 molar ratio (protein:inhibitor) and the protein concentration was adjusted to 0.85 mM in low-salt buffer and crystallization proceeded as described for substrates with A3A-E72A.

### X-ray crystallography
Notwithstanding two distinct crystal habits (tiny flattened needles and thin plates), all structures are approximately isomorphous with space group $P2_1$ ($Z' = 2$) and unit cells of dimensions $a \approx 52$ Å, $b \approx 57$ Å, $c \approx 92$ Å and $\beta \approx 105°$. Data were processed on-site at the Australian Synchrotron using XDS[64,65]. Each structure was solved independently by molecular replacement (MolRep[66]) using the A3A structure PDB ID 5keg[28,67] (space group $I222$) from which metal ions, ssDNA, chloride ions and waters had been stripped. After rigid body refinement with REFMAC5[68] of the CCP4 suite[69], initial electron density maps, visualized with COOT[70], showed clearly the presence, or in one structure absence, of Zn²⁺, along with well-defined electron density for the ssDNA hairpin. Structure elucidation proceeded with rounds of building with COOT and refinement with REFMAC5. In all structures, active site Loop-3 was not well defined, as well as Loop-2 that is remote to the active site. In several structures, phytic acid (inositol hexaphosphate) was ill-defined with only three phosphate groups being well-defined. Supplementary Table S2 presents a summary of crystallographic data, data collection and structure refinement. Supplementary Figs. S1–S4 illustrate crystal packing, molecular structures and superpositions.

### Circular dichroism (CD) spectroscopy of hairpin DNA
CD spectra were recorded using a Chirascan CD spectrophotometer (150 W Xe arc) from Applied Photophysics with a Quantum Northwest TC125 temperature controller. CD spectra (average of at least 3 scans) were recorded between 200 and 350 nm with 1 nm intervals, 120 nm/min scan rate and 10 mm path length followed by subtraction of a background spectrum (buffer only). CD spectra were recorded at ~10 μM DNA concentration in 50 mM Na⁺/K⁺ phosphate buffer, pH 7.0 supplemented with 100 mM NaCl, 1 mM TCEP, 100 μM DSS and 10 % D₂O.

Circular dichroism spectra showed a shift of positive ellipticity from 274 nm for unstructured DNA (T₄CAT) to 286 nm for dC-hairpin which was also accompanied by increase of molar ellipticity (Supplementary Fig. S1d, top panel). Presence of G-C base-pairs in the duplex part of dC-hairpin is evident from four singlets of four imino protons at 13–13.2 ppm in ¹H NMR spectrum (Supplementary Fig. S1d, bottom panel). These data confirm that dC-hairpin is folded in solution and can be used as a scaffold to design A3A inhibitors by using nucleoside-based inhibitors of CDA instead of dC in the loop of dC-hairpin.

### Nuclear magnetic resonance (NMR) spectroscopy and mass spectrometry of hairpin DNA
¹H, ¹³C, ³¹P NMR spectra were recorded on Bruker 500- and 700 MHz spectrometers, the latter with dual-channel cryoprobe. A representative spectrum in the imino region is shown in Supplementary Fig. S1d (top panel). NMR spectra were processed in TopSpin. High-resolution

electrospray mass spectra were recorded on a Thermo Fisher Scientific Q Exactive Focus Hybrid Quadrupole-Orbitrap mass spectrometer. Ions generated by ESI were detected in positive ion mode for small molecules and negative ion mode for oligonucleotides. Total ion count (TIC) was recorded in centroid mode over the $m/z$ range of 100–3000 and analyzed using Thermo Fisher Xcalibur Qual Browser. Mass-spectrometric data on hairpin DNA are presented in Supplementary Table S1.

### Isothermal titration calorimetry (ITC) of interaction of A3A with hairpin DNA

Desalted unmodified TTC-hairpin oligo was purchased (Integrated DNA Technologies) at 1 µmol synthesis scale and dissolved in TE buffer (10 mM Tris/HCl pH 7.9, 1 mM EDTA) to give 10 mM solutions. ITC experiments were conducted at 25 °C using a MicroCal ITC200 (now Malvern Instruments) isothermal titration calorimeter. Protein A3A-E72A, which is a catalytically inactive variant, was dialyzed and diluted with ITC buffer to concentrations of about 50 µM (ITC buffer: 50 mM Na$^+$/K$^+$ phosphate, pH 6.0, 50 mM NaCl, 50 mM choline acetate, 2.5 mM TCEP, 200 µM EDTA with 30 mg/mL bovine serum albumin; after preparation, this buffer was frozen and defrosted before the experiments) and titrated with dC oligonucleotides dialyzed against the above ITC buffer. The concentration ratio of oligonucleotide in the syringe to protein in the cell is generally 10:1 (for 1:1 binding). Supplementary Table S3 presents full analysis of ITC results; Supplementary Fig. S6a, b show the titration curves and derived plots of enthalpy changes *versus* stoichiometry ratio A3A-E72A:hairpin DNA.

### Enzymology of A3A with hairpin substrates and inhibitors

Hairpins as substrates and inhibitors were analyzed as previously described in ref. 43. In short, wildtype A3A was used to compare linear DNA (A$_2$T$_2$**C**A$_4$) and dC-hairpin (T(GC)$_2$TT**C**(GC)$_2$T, bold C is deaminated) at 500 µM in the NMR-based assay (20 °C, pH 7.4, 50 mM Na$^+$/K$^+$ phosphate buffer, supplemented with 100 mM NaCl, 1 mM TCEP, 100 µM sodium trimethylsilylpropanesulfonate (DSS) and 10% D$_2$O; enzyme concentration in assay: 140 nM, from dilution of wildtype A3A ( > 200 µM) in low-salt buffer. The NMR-based assay yields the initial velocity of deamination of various ssDNA substrates, including the modified ones[40], in the presence of A3 enzymes. Consequently, the Michaelis–Menten kinetic model was used to characterize substrates and inhibitors of A3. Moreover, use of dC-containing hairpin as a substrate of A3A allowed us to use a global regression analysis of the kinetic data over the entire time course of the reaction using Lambert's W function (integrated form of the Michaelis-Menten equation).

The course of the reaction was followed by $^1$H NMR until the substrate was consumed (up to 28 h, depending on the experiment performed). Subsequently the amount of substrate or product at each time point was calculated by integrating the decreasing substrate peak at 7.752 ppm (singlet) or the increasing product peak at 5.726 ppm (doublet) and calibrated by the area of DSS standard peak at 0.0 ppm. Using the known concentration of the standard, the peak was converted to a corresponding substrate concentration. The time at which each spectrum was recorded as a difference to the first spectrum was used as the time passed. The product or substrate concentration *versus* the time of reaction was plotted and fitted using the integrated form of the Michaelis-Menten equation:

$$[S]_t = K_m W\left(\frac{S_0}{K_m}\exp\left(\frac{[S]_0 - V_{max}t}{K_m}\right)\right) \qquad (1)$$

where $W$ is Lambert's W function, $[S]_t$ is the substrate concentration at specific time, $[S]_0$ is the initial substrate concentration, $V_{max}$ and $K_m$ are the Michaelis-Menten constants and $t$ is the time. The two Michaelis-Menten constants, $k_{cat}$ and $K_m$, the initial substrate concentration and

an offset which corrects for the integration baseline in the NMR spectra were fitted using Lambert's $W$ function in Gnuplot.

By varying the concentration of an inhibitor, the plots of observed $K_m$ versus inhibitor concentration were obtained, fitted with a linear function ($f(x) = a + b\,x$) and $K_i$ values were calculated as $a/b$, with error propagation as described in ref. 40 (Supplementary Fig. S6c).

### Evaluation of stability of PS-hairpins against enzymatic digestion

Separately TTC-hairpin, PS-TTdZ-hairpin, and 3′-fluorescein-labelled PS-TTdZ-hairpin-FAM and PS-TTdZ-hairpin-FAM (each 15 µM in 50 mM Tris-HCl buffer, 10 mM MgCl$_2$, pH 8.0, 37 °C) were treated with snake venom phosphodiesterase (phosphodiesterase I, Sigma, 32 mU/mL). The percent degradation over time (0–360 min) was monitored by anion-exchange chromatography for the indicated times at 37 °C.

### DNA deaminase activity assays

HEK 293 T (RRID: CVCL_0063) cells (ATCC, USA) were maintained in RPMI-1640 (#SH30027.01, Cytiva, USA) supplemented with 10% fetal bovine serum (#10437028, Gibco, ThermoFisher, USA) at 37 °C with 5% CO$_2$ in a humidified atmosphere. The ssDNA deaminase activity was performed as described in ref. 71. Whole cell lysates were prepared, placed on ice, and immediately used for the deaminase assay. Inhibitor oligos (and controls) were heated to 80 °C for 5 min, and then cooled to RT to induce hairpin formation. They were then prepared at varying concentrations in 5 µL and combined with 10 µL of cell lysate at 37 °C for 15 min to promote binding of oligos to A3A. To this reaction, 5 µL of a mastermix containing 0.25 µL RNAse A, 800 nM fluorescent ssDNA substrate, 10x UDG buffer (NEB #M0280), and 0.25 µL UDG (NEB #M0280) were added to each sample for a total volume of 20 µL and incubated at 37 °C for 1 h. The fluorescently-labeled oligo has the following sequence: (5′-(ATT)$_3$ATT**C**GAATGG(ATTT)$_6$-fluorescein-3′). Reactions were fractionated on a 15% Urea-TBE acrylamide gel, imaged with a Typhoon FLA-7000 imager (GE Healthcare), and then quantified using ImageQuant (Cytiva, USA).

### Cellular uptake and localization of DNA oligomers

MCF7 (RRID: CVCL_0031) cells (ATCC, USA) were maintained in DMEM (Gibco, ThermoFisher Scientific, USA) supplemented with 1% penicillin/streptomycin (Gibco), and 10% fetal bovine serum (Gibco) at 37 °C with 5% CO$_2$ in a humidified atmosphere.

Briefly, MCF7 cells were transfected with FAM-labelled hairpins using X-tremeGENE™ HP DNA Transfection Reagent (Roche, USA; 1.0 µL). After 16 h, the cells were washed twice with phosphate-buffered saline (PBS) containing MgCl$_2$ and CaCl$_2$, fixed in 4% paraformaldehyde/PBS for 15 min at room temperature (RT), and then washed with PBS. Cells were then stained with Hoechst 33342 before imaging on a Zeiss LSM 900 Scanning Confocal Microscope using an oil-immersion 63 objective lens (NA 1.4). Laser excitation wavelengths and collection ranges appropriate to the fluorophores of each sample were used to detect the emission spectra of the specific combination of Hoechst 33342 (excitation at 405 nm, emission monitored at 410–530 nm) and FAM 555 (excitation at 496 nm, emission monitored at 511-579 nm). All images were digitally processed for presentation with ImageJ (Rasband, W. 2014. ImageJ. U.S. National Institutes of Health, Bethesda, MD). Confocal microscopy images of cellular uptake of inhibitors are provided in Fig. 5a and Supplementary Fig. S7.

### MTT cell viability assay

MCF-7 or MDA-MB-453 cells were seeded in 96-well plates at a density of $1.8 \times 10^4$ and $2 \times 10^4$ cells/well in 90 µL complete DMEM and incubated for 24 h to adhere under conditions described in section 1.8. Then, 10 µL of transfection mixture containing Opti-MEM™ I Reduced-Serum Medium (Gibco, Thermo Fisher), the respective DNA hairpins and 0.2 µL of X-tremeGENE™ HP DNA Transfection Reagent (Roche)

were added to cells and incubated for 24 h or 48 h. Cell viability in the absence or presence of hairpins was assessed by MTT assay. Briefly, 10 μL MTT solution (Biotium) was incubated with the cells for 3 h. Formazan crystals formed in living cells were solubilized with 200 μL/well of DMSO. The absorbance was read at the test and reference wavelengths (λ) of 570 and 620 nm, respectively, on a POLARstar Omega plate reader (BMG Labtech). The percentage of living cells was calculated as follows:

$$Viability(\%) = (OD_{570\,exp} - OD_{620\,exp})/(OD_{570cont} - OD_{620cont}) \times 100, \quad (2)$$

where $OD_{570exp}$ and $OD_{570cont}$ correspond to optical density in experimental and control wells, respectively, at λ = 570 nm, and $OD_{620exp}$ and $OD_{620cont}$ correspond to optical density in experimental and control wells, respectively, at λ = 620 nm. Results are shown in Supplementary Fig. S8a.

### Inhibition of A3A activity *in cellulo*

As summarized in Supplementary Fig. S8b, HEK 293 T cells stably transduced with the live-cell deaminase reporter have been reported[60,61]. Semi-confluent cells in a 24-well TPP plate (#Z707791, Millipore Sigma, Merck, DE) were transfected with the following: pcDNA3.1-A3A-HA (20 ng), LTR-gRNA-Cas9n-UGI-NLS-Puro-LTR (400 ng), and varying concentrations of inhibitor (or control) oligos using TransIT-LT1 (Mirus Bio) as the transfection reagent. The plate was imaged using an Incucyte (Sartorius, USA) over the course of 68 h. Live-cell images of orange, green, and phase image channels were captured with a 4x objective every four hours after the initial transfection with five images per well in a fixed grid. mCherry- and GFP-positive cells were identified with internal cellular analysis software. The rolling slope for each concentration between 30 and 50 h was calculated using GraphPad Prism (Dotmatics) software (first derivative between two time points) and represented as an average in Fig. 5c. After 68 h, cells were collected and prepared for immunoblots. Replicate data are provided in Supplementary Fig. S9.

### Immunoblots

Immunoblots were prepared as described in ref. 61. Samples were separated by either a 4–20% Criterion TGX Precast gel (#5671095, Bio-Rad, USA) or 4–15% Mini PROTEAN TGC gel (#4561084, Bio-Rad, USA), and then transferred to nitrocellulose membranes (#1620112, Bio-Rad, USA). Primary antibodies include mouse anti-Tubulin (#T5168, 1:10000, Sigma-Aldrich), rabbit anti-HA (#3724 S, 1:2500, Cell Signaling), and rabbit anti-Cas9 (#ab189380, 1:5000, Abcam). Secondary antibodies used were goat anti-rabbit IRdye800 (LI-COR, #925-32211, 1:10000) and goat anti-mouse IRdye680 (LI-COR, #926-69020, 1:10000). Raw immunoblots are included as Supplementary Fig. S10. The ladder used to mark the molecular weight is the PageRuler Prestained Protein Ladder (10 to 180 kDa, #26616 ThermoFisher Scientific).

### Reporting summary

Further information on research design is available in the Nature Portfolio Reporting Summary linked to this article.

## Data availability

Coordinates and structure factors are available with the following PDB codes: 8FIM (A3A-E72A with TTC-hairpin DNA substrate), 8FIL (Zn$^{2+}$-free A3A-E72A with TTC-hairpin DNA substrate), 8FIK (A3A-E72A with ATTC-hairpin DNA substrate), 8FII (wildtype A3A with TTFdZ-hairpin inhibitor form 2), 8FIJ (wildtype A3A with TTFdZ-hairpin inhibitor form 1). All other data are reported in the main manuscript or supplementary materials.

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

## Acknowledgements

We are grateful to beam-line scientist Dr. Alan Riboldi-Tunnicliffe for assistance with remote access to the Australian Synchrotron during Covid19 lockdown. We thank Dr. Patrick J.B. Edwards and Mr. David Lun for their assistance with use of the NMR and mass spectrometry facilities at Massey University. We thank B. Moriarity for contributing to the mentorship of AER, and H. Aihara, M. Carpenter, D. Harki, and R. Amaro for thoughtful feedback. Financial support of the Health Research Council of New Zealand in partnership with Breast Cancer Cure, and Breast Cancer Foundation NZ (grant 20/1355) and Kiwi Innovation Network with Massey Ventures Limited (grant MU002391), as well as the support of the School of Natural Sciences, Massey University is gratefully acknowledged. Access to the MX2 Beam Line of the Australian Synchrotron was facilitated by the NZ Synchrotron Group Ltd financially supported by the Ministry of Business Innovation and Enterprise (MBIE) and a consortium of New Zealand universities, including Massey University. Cancer studies in the Harris lab are supported by NCI P01-CA234228 and a Recruitment of Established Investigators Award from the Cancer Prevention and Research Institute of Texas (CPRIT RR220053). RSH is an Investigator of the Howard Hughes Medical Institute, a CPRIT Scholar, and the Ewing Halsell President's Council Distinguished Chair at University of Texas Health San Antonio.

## Author contributions

V.V.F., E.H., T.K.H. and R.S.H. designed experiments, supervised the research and analyzed data; S.H., H.M.K., Y.S., G.B.J., M.B., J.F., T.K.H. and A.E.R. performed experiments; G.B.J. wrote the first draft and V.V.F., E.H., T.K.H., A.E.R. and R.S.H. added sections and edited the manuscript.

## Competing interests

The authors declare no competing interests.
