## [Peer Review File · Nature Communications]

Structure-guided inhibition of the cancer DNA-mutating enzyme APOBEC3AREVIEWER COMMENTS

Reviewer #1 (Remarks to the Author):

Stefan and his colleagues did structure-guided inhibitor design against DNA deaminase A3A in this manuscript. They determined several X-ray structures of FdZ contained hairpin dsDNA with A3A or its variant. Then, they tested anti-cancer activities of PS derivatives of these inhibitors against A3A in cells. The results were solid, noteworthy, and very interesting for the researches in the APOBEC field. However, due to that some figures and detailed description of the experiments were not accessible, the conclusion should be supported by adding a little more data. Therefore, before it is accepted, the authors should make major revisions.

- (1) The references about the reported X-ray and NMR structures of A3A in complex with linear DNA were not fully cited; the author should not only cite 3 reported papers (ref. 18-19 and 23 in original manuscript) which looked important or were done by themselves in the revised manuscript.
- (2) In TTFdZ-inhibitor, the sequence in the part of dsDNA has any effects on the inhibitor activities? The authors should do some changes in the sequence, or add some analysis about the results from the complex structure in the manuscript.
- (3) What roles of 5-fluoro were in inhibitor design? The author should describe why they introduce fluorine atom into loop inhibitor in the revised manuscript, although it might be described in previous paper or in authors' published work.
- (4) R28 and H29 are important for identification of the base T₂ in the inhibitor TTFdZ contained dsDNA, the authors should make some mutations to test the binding affinity changes of the inhibitor DNA to A3A or its mutants.
- (5) It was very skillful to test in vivo anti-cancer activities using PS-TTFdZ hairpin dsDNA. Did the authors do PS modification in all normal O-P-O backbone in the hairpin contained dsDNA, all just in the part of the hairpin? The authors should describe it very clearly in the revised manuscript, by presenting the MS spectra of these PS modified oligos in the supporting information.
- (6) In table S1, there were no MS data at all. Thus, in section of methods, NMR and MS part, the authors should make some revisions.
- (7) In section of ITC in methods, the authors mentioned Table S3, figures S12 and S13, where were they? I did not find them. Figures S11, S14-18 were either not found. In supplementary file, only figure S1 and tables S1 and S2 were provided. The authors should carefully check and present all figures, tables in the revised manuscript.
- (8) As a special motif, TTC could also be identified by other DNA deaminases. How to avoid off-target possibilities of the inhibitor in the future when these PS-modified hairpin DNA is used as anti-cancer agent in human? Add some discussion in the conclusion part or other places in the revised

manuscript.

Reviewer #2 (Remarks to the Author):

The manuscript by Harjes et al reports the structure of the DNA mutator enzyme A3A bound to substrate or inhibitory DNA hairpins. A3A is both an antiviral enzyme and driver of tumor diversity -by editing cellular DNA- through its cytosine deaminase activity. This study shows for the first time that A3A editing on genomic DNA in cells can be potently inhibited by DNA hairpins containing a cytosine transition state analog 5' fluoro zebularine. The work is a tour de force combining 6 high-resolution crystal structures in concert with enzyme kinetic and direct binding studies to explain why hairpin inhibitors containing T-T-FZ , where FZ is fluorozebularine, are more potent than linear DNA. The structural , kinetic and direct binding studies suggest the hairpin pre-organizes the substrate for efficient catalysis and that specific amino acids (R28 and H29) specifically interact with the hairpin loop to enable tight binding. They beautifully explain how H29 of A3A acts DNA base mimic, base pairing with a nucleotide at position -2 and explaining the specificity of A3A for a pyrimidine at this position. The authors show that conversion of the DNA phosphate backbone from phosphodiester to phosphorothioate linkages protects DNA from phosphodiesterase (and cellular exonucleases) without significantly compromising inhibition, allowing them to test phosphorothioate hairpin DNA inhibition of A3A in cells. They show that hairpin DNA inhibitors enter the nucleus and inhibit the activity of A3A when fused to Cas9.

The manuscript is significant because it explains the structural preference of A3A activity on DNA hairpins which are physiologically and pathologically relevant in cancer evolution . In addition, compounds that inhibit A3A could be important for dissecting its mechanism (as tool compounds) or as therapeutics. The quality of the work and insights provided make it suitable for publication in Nature Communications.

Minor comments that should be addressed:

1-P.3 line 95. Define semi-salt bridge.

2-Fig 1f, the filled red-ellipse around H29 and R28 obfuscate the interactions depicted

3-P.4 line 130, referring to H29 "This provides an additional base-pair in which one base is from DNA and another is from protein" H29 is an amino acid, not a base, suggest re-writing to indicate protein amino acid mimics DNA base, a common strategy in nucleic acid binding

proteins.

4-Fig 1 legend, p. 12 , line 417

G-4 is a typo, should be C-4.

5-p 5 line 137, "peptide N" should read "peptide backbone N".

6-p 5 line 147, "A3 enzymes prefer DNA over RNA structures" re-write as "A3 active sites prefer DNA over RNA structures". Some A3s bind RNA quite potently using the pseudo catalytic domain (e.g. A3G).

7-p 6, line 171 cites k_{cat}/K_m comparison. Please make a call to the relevant table or figure.

Responses to referees' reports - NCOMMS-23-10152-T

Referee 1:

Stefan and his colleagues did structure-guided inhibitor design against DNA deaminase A3A in this manuscript. They determined several X-ray structures of FdZ contained hairpin dsDNA with A3A or its variant. Then, they tested anti-cancer activities of PS derivatives of these inhibitors against A3A in cells. The results were solid, noteworthy, and very interesting for the researches in the APOBEC field. However, due to that some figures and detailed description of the experiments were not accessible, the conclusion should be supported by adding a little more data. Therefore, before it is accepted, the authors should make major revisions.

Response: We thank referee 1 for appreciating that the “The results were solid, noteworthy, and very interesting ...”. We apologize for having supplementary materials that were difficult to access, and we believe we have now corrected the problem. The answers to your queries can be found in the revised manuscript or in the supplementary materials, as detailed in additional responses below.

Referee 1 continues:

(1) The references about the reported X-ray and NMR structures of A3A in complex with linear DNA were not fully cited; the author should not only cite 3 reported papers (ref. 18-19 and 23 in original manuscript) which looked important or were done by themselves in the revised manuscript.

Response: We apologise for this omission of the NMR structure of A3A-E72A with substrate by Liu *et al.* We now reference this NMR structure, along with changes to the text, which now reads (lines 54-63):

“A3A exhibits an intrinsic preference for deamination of cytosine bases within 5'-YTCD motifs, where Y denotes C or T and D is A, G, or T^{15-17,21-26}. The strong preference for TC dinucleotide motifs is explained by previous crystal and NMR structures of linear ssDNA bound to an inactive mutant of A3A, which revealed the atomic contacts (3 H-bonds) with thymine (-1 T) as well as the binding pocket for the target cytosine²⁷⁻³³. However, preferences of A3A for nucleotides flanking the TC-sequence are as-yet-unexplained. Crystallographic studies also revealed that upon binding to A3A the ssDNA adopts a distinctive U-shape that projects cytosine into the A3A active site²⁷⁻²⁸. Consistent with this observation, DNA hairpins with loops of three and four nucleotides have been shown to be more potent substrates of A3A in comparison to linear analogs^{16,34-38}.”

Referee 1 continues:

(2) In TTFdZ-inhibitor, the sequence in the part of dsDNA has any effects on the inhibitor activities? The authors should do some changes in the sequence, or add some analysis about the results from the complex structure in the manuscript.

Response: As recommended, we have added additional text in the **Results and Discussion** sections, which expands our analysis (we do present structures with different stems with AT pairs), but does not alter our interpretations, of the multiple different structures that we have

determined. We also discuss the work of Rahul Kohli's lab (PMID: 36475588), where the variation of composition of the stem of inhibitors had negligible effect on inhibition. Therefore, we strongly feel that additional structures and enzymatic data are beyond the scope of the present studies.

We have added the following text pertinent to stem composition (lines 177-182):

"The close superposition of the stems, which include in several structures two AT pairs, along with GC pairs, and the lack of specific interaction of most of the DNA stem with the protein (**Fig. 2b, c; Supplementary Fig. S4a**), suggests that reactivity of dC-containing hairpins and inhibition by **FdZ-** (or **dZ-** or 5-methyl-dZ-) hairpins is unrelated to stem composition of DNA hairpins, where substrate or inhibitor moiety is located at the 3' end of the loop. This is consistent with the recent observation that variation of stem composition had negligible effect on inhibition⁴⁴."

Referee 1 continues:

(3) What roles of 5-fluoro were in inhibitor design? The author should describe why they introduce fluorine atom into loop inhibitor in the revised manuscript, although it might be described in previous paper or in authors' published work.

Response: As recommended, we have added text to describe the rationale behind the fluorine atom (lines 83-88):

"The mechanism for CDA deamination was proposed based on several crystal structures in which zebularine and 5-fluorozebularine accept a Zn^{2+} -bound water molecule across the N3-C4 double bond forming a tetrahedral intermediate in complex with the enzyme (**Fig. 1a**)⁴⁶⁻⁴⁸. Accordingly, our prior work with A3A has indicated that, of these two cytosine analogs, the latter fluoro-containing nucleobase is a more potent inhibitor in the context of ssDNA⁴¹."

Referee 1 continues:

(4) R28 and H29 are important for identification of the base T-2 in the inhibitor TTFdZ contained dsDNA, the authors should make some mutations to test the binding affinity changes of the inhibitor DNA to A3A or its mutants.

Response: We thank the reviewer for suggesting this experiment and we have revised our text to refer to prior literature where these amino acids have been mutated (lines 201-211):

"The importance of Arg28 and His29 to recognition of T⁻² is shown also by the following results. For A3A on mutation of Arg28 to alanine, diminished activity towards linear ssDNA was reported²⁷. Moreover, mutation of His29 to arginine in A3A (H29R) caused a 10-fold diminution of activity against a linear ssDNA compared to wildtype A3A⁵³⁻⁵⁴. Based on co-crystal structures here, we predict that the longer side chain of Arg in the H29R mutant will place the polar head group in a suboptimal position beyond the reach of O2 of T⁻² and with poor π -stacking with the nucleobase at +1. Interestingly, the nearly identical (>90%) A3B catalytic domain lacks a corresponding histidine in its loop 1 region (it naturally has an arginine), and A3B shows no preference for pyrimidines in the -2 position of linear ssDNA substrates with target C at position 0^{16,55}. As yet, there is no structural information on wildtype A3B_{CTD} with loop 1 in an open conformation with substrate or inhibitor bound to test this supposition."

Referee 1 continues:

(5) It was very skillful to test in vivo anti-cancer activities using PSTTFdZ hairpin dsDNA. Did the authors do PS modification in all normal OPO-O backbone in the hairpin contained dsDNA, all just in the part of the hairpin? The authors should describe it very clearly in the revised manuscript, by presenting the MS spectra of these PS modified oligos in the supporting information.

Response: All phosphate linkers were replaced by phosphorothioates. The sequences and mass-spectrometric data on normal phosphate-linked oligos as well as on their phosphorothioated analogs are provided in **Supplementary Table S1**.

Referee 1 continues:

(6) In table S1, there were no MS data at all. Thus, in section of methods, NMR and MS part, the authors should make some revisions.

Response: The data were present but in a mis-labelled table. We apologies for the confusion caused. The data are now in **Supplementary Table S1**.

Referee 1 continues:

(7) In section of ITC in methods, the authors mentioned Table S3, figures S12 and S13, where were they? I did not find them. Figures S11, S14-18 were either not found. In supplementary file, only figure S1 and tables S1 and S2 were provided. The authors should carefully check and present all figures, tables in the revised manuscript.

Response: Again, apologies for the confusion. ITC data appeared in Extended Data Figure 4 and in Supplementary Information Table S1. Similarly, the cellular localisation measurements should have been referenced as Extended Data Figure 5. They now are present in **Supplementary Fig. S6** and **Supplementary Table S3** and **Supplementary Fig. S7**.

Referee 1 continues:

(8) As a special motif, TTC could also be identified by other DNA deaminases. How to avoid off-target possibilities of the inhibitor in the future when these PS-modified hairpin DNA is used as anti-cancer agent in human? Add some discussion in the conclusion part or other places in the revised manuscript.

Response: This is a fair point. To address it, we have included the following sentence in our revised Discussion (lines 283-288): "Although we have not specifically tested the hairpin inhibitors described here against related human enzymes, the most similar TC-preferring A3 family member (A3B) is not known to prefer hairpin substrates^{16,54,61} and other TC-preferring A3s have yet to be tested systematically with hairpins. It is unlikely that A3G and AID, with non-TC preferences, would be inhibited by the hairpins described here. However, when hairpin or other A3A inhibitors get closer to clinical development, these and other off-target possibilities should be examined in dedicated biochemical experiments."

And beginning at line 289 we conclude with:

“A structural understanding of the hairpin preference of A3A helped inform the design of substrate-mimicking FdZ inhibitors. Importantly, phosphorothioated derivatives are resistant to nuclease degradation and can be directed to the nucleus with the aid of commonly used transfection reagents. Moreover, we have obtained an important proof of concept here through the inhibition of the mutagenic activity of A3A in living cells with a PS-FdZ hairpin. Further optimization of such inhibitors may lead to small molecules that can be used in a therapeutic setting to slow rates of tumor evolution and improve clinical outcomes for patients with A3A-driven tumors.”

Response to Referee 2:

The manuscript by Harjes et al reports the structure of the DNA mutator enzyme A3A bound to substrate or inhibitory DNA hairpins. A3A is both an antiviral enzyme and driver of tumor diversity - by editing cellular DNA- through its cytosine deaminase activity. This study shows for the first time that A3A editing on genomic DNA in cells can be potently inhibited by DNA hairpins containing a cytosine transition state analog 5' fluoro zebularine. The work is a tour de force combining 6 high-resolution crystal structures in concert with enzyme kinetic and direct binding studies to explain why hairpin inhibitors containing T-T-FZ, where FZ is fluorozebularine, are more potent than linear DNA. The structural, kinetic and direct binding studies suggest the hairpin pre-organizes the substrate for efficient catalysis and that specific amino acids (R28 and H29) specifically interact with the hairpin loop to enable tight binding. They beautifully explain how H29 of A3A acts DNA base mimic, base pairing with a nucleotide at position -2 and explaining the specificity of A3A for a pyrimidine at this position. The authors show that conversion of the DNA phosphate backbone from phosphodiester to phosphorothioate linkages protects DNA from phosphodiesterase (and cellular exonucleases) without significantly compromising inhibition, allowing them to test phosphorothioate hairpin DNA inhibition of A3A in cells. They show that hairpin DNA inhibitors enter the nucleus and inhibit the activity of A3A when fused to Cas9.

The manuscript is significant because it explains the structural preference of A3A activity on DNA hairpins which are physiologically and pathologically relevant in cancer evolution. In addition, compounds that inhibit A3A could be important for dissecting its mechanism (as tool compounds) or as therapeutics. The quality of the work and insights provided make it suitable for publication in Nature Communications.

We acknowledge Referee 2's very positive comments.

First an oversight to address: Referee 2 stated that "...inhibit the activity of A3A when fused to Cas9." To ensure fair competition between gDNA and the hairpin inhibitor for the affections of A3A, A3A was untethered from Cas9n. We checked the manuscript to ensure clarity on this point. The figure also emphasises this point.

Minor comments that should be addressed:

1-P.3 line 95. Define semi-salt bridge.

Response: This is between a neutral moiety and a charged moiety e.g. Glu/Asp-COO⁻...H₂N-C(=O)-Asn/Gln. We thought this was a well-known term, but we have eliminated its mention to preserve simplicity (and yellow-highlighted the clarification at lines 111-113).

"The catalytic glutamic acid residue, Glu72, which functions as a general acid/base, hydrogen bonds to C4-OH and N3-H of hydrated FdZ. We expect for such interactions that Glu72 is present in the carboxylate form in the crystal structure."

Referee 2 continues:

2-Fig 1f, the filled red-ellipse around H29 and R28 obfuscate the interactions depicted.

Response: We have removed the shading, and indeed repackaged the figures into more digestible ones with fewer parts to each. This is now Fig. 2c.

Referee 2 continues:

3-P.4 line 130, referring to H29 "This provides an additional base-pair in which one base is from DNA and another is from protein" H29 is an amino acid, not a base, suggest re-writing to indicate protein amino acid mimics DNA base, a common strategy in nucleic acid binding proteins.

Response: We have clarified, as suggested, so that lines 198-200 now read as follows:

"Therefore, this quasi-base pair between pyrimidine at position –2 and His29 is able to stack on top of the first CG base pair (at positions –3 or –4 and +1) at the head of the stem (**Fig. 3d**)."

We have expanded this section, breaking into two parts to explain more fully the structural basis of preference for a pyrimidine at position –2 and for A, G, or T at +1 but rarely C. This includes a new frame in Fig. 3 (**Fig. 3f**) where we have modelled the van der Waals interactions of the thymine methyl group with Ala59-CB and Lys30-CG/CE, and the hydrogen bond between Lys30-NZ and the carbonyl at C4 on thymine (lines 214-228).

"Cytosine at +1 is rarely observed in the A3A-induced mutation spectra in model systems^{16,17,24,2} and it is also strongly underrepresented in the overall A3 mutation signature in tumors^{55,56}. At least for hairpin structures, π - π stacking of His29 with the six-membered ring of G⁺¹ (or alternatively A⁺¹) and van der Waals interaction of the CH₂ group (C β) of His29 with the five-membered ring of G⁺¹ (or alternatively A⁺¹) explains the preference for purines over pyrimidines in the +1 position (**Fig. 3d, e**). The preference for thymine over cytosine is more subtle. In part, cytosine lacks the electron-donating methyl group of thymine, which leads to less favorable π - π interactions with His29. In addition, modelling T at position +1 and adjusting the Lys30 side chain to a favoured conformation reveals a small hydrophobic pocket that brings the methyl groups of T⁺¹ and Ala59 and the methylene groups C γ and C ϵ of Lys30 into van der Waals contact. Moreover, the terminal amino group of Lys30 hydrogen-bonds with the carbonyl moiety at C4 of T⁺¹ (**Fig. 3f**); cytosine lacks this carbonyl, instead having an amino group. In this context, we also note that *in vitro* A3A prefers to deaminate the highlighted C of a suboptimal linear 5'-ATTCCCAATT substrate, whereas A3B_{CTD} attacks the 5'-most C^{40,54}. Cytosine lacks this methyl group and carbonyl group and thus substrates presenting YTTCD (D = A, G, T) to A3A are favoured over those presenting YTTCC."

Referee 2 continues:

4-Fig 1 legend, p. 12 , line 417. G-4 is a typo, should be C-4.

Response: Corrected – many thanks for eagle eyes. Now at in caption for **Fig. 3d** at line 780.

Referee 2 continues:

5-p 5 line 137, "peptide N" should read "peptide backbone N".

Response: We have addressed this point on lines 196-198, with addition of an H to the N as: "...In addition, the water molecule that bridges the carbonyl O2 of T⁻² (and potentially also of C at –2) to the peptide backbone NH of His29 cannot be accommodated for purine at –2 (**Fig. 3d**, **Supplementary Fig. S5b, d**)."

Referee 2 continues:

6-p 5 line 147, "A3 enzymes prefer DNA over RNA structures" re-write as A3 active sites prefer DNA over RNA structures". Some A3s bind RNA quite potently using the pseudo catalytic domain (e.g. A3G).

Response: Happy to make this clarification. Rewritten, omitting the word "structures" on lines 93-94 as "A3 enzymes in general prefer as substrates DNA over RNA and, in accordance, all deoxyribose moieties..."

Referee 2 continues:

7-p 6, line 171 cites k_{cat}/K_m comparison. Please make a call to the relevant table or figure.

Response: Done. Lines 241-242 read as "...if k_{cat}/K_m are compared at pH 7.4 (Supplementary Table S4a)"

REVIEWERS' COMMENTS

Reviewer #1 (Remarks to the Author):

No more comments from my side. The research in this manuscript was important and interesting to the A3 field. The evidences were solid to support the authors' conclusion. The techniques used in this report were used correctly.

Chunyang Cao